# Metagenomic and taxonomic profiling of phyllosphere bacteria from *Mangifera indica* in response to urban air pollutants in Medellín, Colombia

Natalia Bernal Hernández[1], Héctor Alejandro Rodríguez Cabal[2,3]*, Nancy J. Pino[1], Sara Ramírez Restrepo[4], Luisa María Múnera Porras[5]*

1 Diagnostic and Pollution Control Research Group (GDCON), School of Microbiology, University of Antioquia, Medellín, Colombia, 2 Grupo en AgroBiotecnología, Instituto de Biología, Facultad de Ciencias Exactas y Naturales, Universidad de Antioquia, Medellín, Colombia, 3 Grupo de Biotecnología Vegetal, Facultad de Ciencias Agrarias, Universidad Nacional de Colombia Sede Medellín, Medellín, Colombia, 4 Biosciences Research Group, Faculty of Health Sciences, Colegio Mayor de Antioquia, Medellín, Colombia, 5 Research Group on Health and Sustainability, School of Microbiology, University of Antioquia, Medellín, Colombia

* luisam.munera@udea.edu.co (LMMP); hearodriguezca@unal.edu.co (HARC)

## Abstract

Urban trees and their phyllosphere-associated microbiota constitute a promising nature-based solution for mitigating urban air pollution. In this study, we characterized the taxonomic composition, diversity patterns, and functional potential of bacterial communities inhabiting the phyllosphere of *Mangifera indica* in two urban sites of Medellín, Colombia, with contrasting pollution levels and across two time points, analyzing a total of 12 samples. We integrated 16S rRNA gene amplicon sequencing, performed on the Illumina MiSeq platform, with shotgun metagenomic sequencing generated on the Illumina NovaSeq 6000 platform to assess community structure and the presence of genes involved in the degradation of airborne organic pollutants. Bacterial assemblages were dominated by *Pseudomonadota (Proteobacteria)*, *Actinomycetota*, and *Bacteroidota,* with genera such as *Methylobacterium, Pseudomonas*, and *Serratia* consistently prevalent. Alpha diversity was higher in the highly polluted downtown, while beta diversity was shaped primarily by temporal variation. Functional annotation of metagenome-assembled genomes (MAGs) uncovered genes encoding complete aromatic hydrocarbon degradation pathways, including naphthalene, toluene, xylenes, and benzoate. Both ortho- and meta-cleavage routes for catechol degradation were detected, with temporal shifts in pathway dominance linked to changes in the abundance of key degraders taxa. These results reflect genetic potential for xenobiotic degradation within the *M. indica* phyllosphere microbiota, modulated by environmental conditions. Our findings highlight the ecological role of phyllosphere bacteria as contributors of inferred functional capacity relevant to

**Data availability statement:** Il relevant data are within the manuscript and its Supporting Information files.

**Funding:** This project was funded by the Ministry of Science, Technology and Innovation of Colombia (Minciencias) under Call 918-2022: Strengthening Regional Research Capacities in Public Health. Minciencias had no role in the study design, data collection and analysis, decision to publish, or preparation of the manuscript. The entity's URL is: https://minciencias.gov.co/.

**Competing interests:** The authors have declared that no competing interests exist.

atmospheric bioremediation and supports their integration into microbiome-informed green infrastructure strategies.

## Introduction

Air pollution is a critical environmental and public health issue, particularly in urban areas of the Global South. According to the World Health Organization (WHO), over 99% of the global population lives in areas where air quality fails to meet safety standards [1]. In 2021, air pollution was responsible for approximately 8.1 million deaths globally, about one in every eight, ranking as the second leading risk factor for premature mortality globally, after high blood pressure. It also emerged as the second most significant risk factor for deaths in children under five years of age, following malnutrition [2].

In Colombia, poor air quality is estimated to cause over 8,000 deaths annually, longside significant economic losses [3,4]. Regionally, the Aburrá Valley, particularly the city of Medellín, experiences recurrent air pollution episodes due to a combination of rapid urbanization, vehicular emissions, topographical constraints, and meteorological conditions that hinder pollutant dispersion [5,6].

Nature-based solutions (NbS), such as the strategic deployment of urban vegetation, are increasingly promoted to mitigate the impacts of air pollution [7]. Trees not only capture airborne particulate matter but also contribute to pollutant degradation through complex interactions with their phyllosphere microbiota [8]. *Mangifera indica*, commonly found in Medellín's green corridors, is particularly relevant due to its high tolerance to air pollutants and its structural capacity to retain airborne particles [9,10]. Its evergreen nature, high leaf surface area, and waxy cuticle make *M. indica* an ideal model for assessing long-term interactions between phyllosphere microbial communities and urban air pollution [11].

The phyllosphere, defined as the aerial parts of plants, predominantly leaves, hosts bacterial densities ranging from $10^6$ to $10^8$ cells·cm$^{-2}$ [12]. These microorganisms are exposed to intense environmental stressors, including UV radiation, nutrient scarcity, desiccation, and fluctuating pollutant loads. Such selective pressures have been shown to stimulate the acquisition of adaptive traits, including biosurfactant production and the expression of metabolic pathways involved in the degradation of airborne organic compounds [13–15]. Importantly, recent metagenomic studies have identified key genes associated with the degradation of aromatic hydrocarbons (e.g., BTEX—benzene, toluene, ethylbenzene, and xylene—and PAHs—polycyclic aromatic hydrocarbons) in phyllosphere communities of species such as *Magnolia grandiflora, Cedrus deodara, Melia azedarach* and *Carpinus betulus* in industrial and urban settings [16–18]. Urban air pollutants not only deposit on leaf surfaces but can also alter the composition, diversity, and metabolic potential of phyllosphere microbiota, potentially affecting their ability to perform key ecological functions such as pollutant degradation and plant health promotion [19].

Emerging evidence suggests that phyllosphere bacteria may serve as effective bioindicators of urban air quality and hold potential as agents of passive

bioremediation [20,21]. For instance, *Curtobacterium*, *Sphingomonas*, and *Pseudomonas* species have been shown to degrade naphthalene and toluene in planta under urban pollution stress [22,23]. However, these findings remain geographically biased toward temperate or subtropical regions, and functional insights into phyllosphere microbial communities in tropical urban environments remain scarce. Moreover, while 16S rRNA amplicon sequencing is widely used to assess taxonomic composition, comparative metagenomic approaches, such as those applied in sugarcane phyllosphere, reveal broader microbial diversity, functional gene repertoires, and cross-domain interactions, underscoring the value of functional profiling in phyllosphere research [24].

To date, no studies have investigated the taxonomic and functional composition of phyllosphere microbiota in *M. indica* under tropical urban pollution gradients. This knowledge gap is particularly limiting in cities such as Medellín, where BTEX concentrations and particulate matter often exceed regulatory thresholds [25]. Addressing this gap is essential to evaluate the ecological role and biotechnological potential of phyllosphere bacteria in mitigating urban air pollution. Insights from this work may be applicable to other tropical urban centers facing similar air quality challenges, contributing to the global development of microbe-based strategies for air pollution mitigation [15].

This study aims to characterize the bacterial communities inhabiting the phyllosphere of *M. indica* in two urban sites of Medellín with contrasting pollution levels. Specifically, we assess: (i) the taxonomic composition of the bacterial communities using 16S rRNA gene metabarcoding, (ii) their functional gene profiles using shotgun metagenomics, and (iii) the presence of metabolic pathways involved in the degradation of aromatic hydrocarbons. These findings may contribute to the development of microbe-assisted NbS strategies for controlling air pollution in tropical urban settings.

## Materials and methods

Fieldwork was conducted under the General Permit for the Collection of Specimens of Wild Species of Biological Diversity for Non-Commercial Scientific Research, issued by the National Authority of Environmental Licenses (Autoridad Nacional de Licencias Ambientales, ANLA) of the Republic of Colombia through Resolution 1566 of July 24, 2024. This permit authorizes the collection of biological material within the national territory exclusively for scientific purposes and without commercial intent. All sampling and handling activities were covered under this general permit; therefore, no additional permits from regional or local authorities were required."

### Sampling

Sampling was performed at two time points: December 2023 (S1, first sampling) and April 2024 (S2, second sampling), corresponding to the periods before and after the first air pollution episode of the year in the Aburrá Valley (5), At each time point, triplicate samples (n = 3) of 50 g of mature, healthy, fully expanded *Mangifera indica* leaves were collected from two urban sites in Medellín, Colombia, characterized by contrasting air pollution levels: the downtown (DT), representing high pollution exposure (6°14'34.8"N, 75°34'28.5"W), and the southern site (SS), representing lower pollution levels (6°11'39.6"N, 75°34'46.7"W). In total, 12 samples were analyzed: three samples per urban site at each sampling period.

At each site and sampling period, leaves were selected from a single adult, unmanaged tree (i.e., not subject to prunning or irrigation) and randomly collected from the outer crown at mid-to-upper canopy height (2.5–3.5 m). Only intact leaves without visible damage or disease symptoms were included. For each technical replicate, a composite sample was formed by pooling leaves from 3–4 different branches of the same tree to reduce intra-individual variability and better capture the representative phyllospheric microbiota under ambient exposure conditions. This sampling design is consistent with previous phyllosphere microbiome studies that employed repeated sampling of the same host individual to assess temporal microbial dynamics while minimizing confounding inter-individual variation [26–28]. Accordingly, and following recommendations for non-independent microbiome measurements, the data were analyzed as repeated measures to account for intra-individual correlation [29].

Leaves were detached using gardening scissors disinfected with 70% ethanol between trees. All leaf handling was performed using sterile nitrile gloves, Samples were placed in sterile polyethylene bags and were sealed immediately after sample collection to minimize external contamination. Finally, samples were stored in coolers at 4–8 °C during transport, and subsequently frozen at −20 °C until processing.

Meteorological and pollutant data corresponded to monthly averages for each sampling period, calculated from continuous measurements at the nearest SIATA monitoring station. Additionally, average monthly BTEX concentrations were obtained directly from SIATA based on their most recent passive sampling campaign conducted in 2022, after which the agency prioritized the monitoring of other physicochemical variables. All data correspond to the monitoring stations closest to the sampling sites (S1 Fig in S1 File).

## DNA extraction

Bacterial cells were detached from the leaf surface by washing 50 g of leaves in 100 mL of sterile phosphate-buffered saline (PBS, pH 7.4) with gentle shaking at 200 rpm for 10 minutes at ~24 °C. All steps were performed inside a laminar-flow cabinet using only sterile, autoclaved materials to minimize the environmental contamination. The resulting suspension was transferred to 50 mL Falcon tubes and centrifuged twice at 6,000 rpm for 15 minutes at 4 °C to concentrate the bacterial biomass. The resulting pellets were pooled and stored at −20 °C until DNA extraction.

Genomic DNA was extracted from the bacterial pellets using new, single-use QIAamp PowerFecal Pro Kit (QIAGEN, Germany), following the manufacturer's instructions. This kit was selected due to its proven efficiency in recovering microbial DNA from environmental samples with potential inhibitors. DNA concentration and purity were initially assessed using a NanoDrop™ OneC spectrophotometer (Thermo Scientific™, USA). To ensure reliability, DNA integrity was verified via 1% agarose gel electrophoresis. DNA yields ranged from 35.0 to 135.4 ng/µL, with A260/280 ratios between 1.8 and 2.0. Extraction blanks consisting of sterile PBS processed identically to the samples were included as negative controls to monitor potential contamination during sample handling and DNA extraction.

## Taxonomic characterization of phyllosphere bacteria

For taxonomic profiling, 16S rRNA gene amplicon sequencing was performed by Macrogen Inc. (Seoul, South Korea) using the Illumina MiSeq platform with a paired-end (2 × 300 bp) protocol and an expected depth of ~500,000 reads. The V3–V4 hypervariable regions were amplified using primers 341F (CCTACGGGNGGCWGCAG) and 805R (GACTACH-VGGGTATCTAATCC). Each sample was sequenced individually and uniquely barcoded to allow multiplexed library preparation.

Raw sequence data were processed using QIIME2 v2024.2 [30]. Denoising and chimera removal were performed using DADA2 [31] with quality trimming based on Phred scores >30. Paired-end reads were merged using the QIIME2 VSEARCH join-pairs method with an average overlap of ~140 bp. The resulting amplicon sequence variants (ASVs) were taxonomically assigned using VSEARCH [32] against the SILVA v138 reference Database [33] using a 99% identity threshold.

QIIME2 artifacts were imported into R v4.4.1 [34] for further analysis using the packages *phyloseq* v1.48.0 [35] and *qiime2R* v0.99.6 [36]. ASVs classified as mitochondria, chloroplasts, or unassigned at the kingdom level were removed. Additionally, low-abundance ASVs representing <0.005% of total reads were filtered out to reduce noise and potential contaminants. To control for environmental and reagent-derived contamination, negative controls (PBS blanks) were sequenced and analyzed in parallel. ASVs detected exclusively in negative controls were subtracted from the dataset using prevalence-based filtering.

Alpha diversity indices (Shannon, Simpson, and Chao1) were calculated based on rarefied ASV counts, using a standardized depth of 31,985 reads per sample. This rarefaction depth was selected based on the sample with the lowest sequencing depth after quality filtering to maximize sample retention while minimizing data loss. Statistical differences between groups were evaluated using ANOVA. Beta diversity was assessed using Principal Coordinates Analysis (PCoA)

based on Bray–Curtis dissimilarities, and differences in community composition were tested using permutational multivariate analysis of variance (PERMANOVA, 999 permutations), and homogeneity of multivariate dispersion (PERMDISP, 999 permutations) using the vegan package v2.6-8 [37]. Taxonomic composition was visualized with bar plots generated using ggplot2 v3.5.1 [38].

## Shotgun metagenomic sequencing and sequence processing

Shotgun metagenomic sequencing was conducted by Macrogen Inc. (Seoul, South Korea) using the Illumina NovaSeq 6000 platform with a paired-end (2 x 150 bp) protocol and an expected depth of ~33M reads. Libraries were prepared using the TruSeq DNA Nano kit (Illumina, USA), with average insert sizes of ~350 bp. DNA concentrations were normalized across all samples prior to sequencing to ensure comparability of sequencing depth. Raw metagenomic data have been deposited in the NCBI Sequence Read Archive (SRA) under BioProject accession number PRJNA1269120.

Quality control of raw reads was performed using FastQC v0.11.9 [39] and MultiQC v1.14 [40]. Adapter trimming and quality filtering were carried out using Trimmomatic v0.39 [41], applying a sliding window of 4 bp and a minimum Phred score of 30. Reads shorter than 120 bp were discarded. Host-associated sequences (chloroplast, mitochondrial, and plant DNA) were filtered out by mapping against the *Mangifera indica* reference genome (NCBI RefSeq GCF_011075055.1) using Bowtie2 v2.4.5 [42].

High-quality, non-host reads were assembled using SPAdes v3.15.5 in metaSPAdes mode [43,44]. Assembly quality was evaluated using MetaQUAST v5.2.0 [45]. Scaffolds were binned using the metaWRAP v1.3.0 pipeline [46], which integrates multiple binning algorithms: MaxBin2 v2.2.6, MetaBAT2 v2.12.1, and CONCOCT v1.0.0 [47–49], followed by bin refinement.

Binning results were filtered to retain only metagenome-assembled genomes (MAGs) with ≥50% completeness and <10% contamination, as assessed using CheckM v1.0.12 [50]. Taxonomic classification of MAGs was performed using GTDB-Tk v2.3.2 with the GTDB R207 database within the KBase platform [51,52]. In addition, taxonomic annotation was cross-validated using MegaBLAST searches against the NCBI nt database [53] to increase classification confidence.

## Functional characterization of phyllosphere bacteria

Functional annotation of the bacterial metagenomes was performed using the metaWRAP v1.3.0 pipeline [46]. Gene prediction and annotation were carried out on each bin using PROKKA v1.11 [54], employing bacterial-specific databases and default parameters. Annotated protein-coding sequences were subsequently mapped to KEGG Orthology (KO) identifiers using GhostKOALA [55] with the KEGG GENES database (release 107.0, April 2024). Functional assignments were retained only when meeting the default KO confidence threshold (bit score > 60).

KEGG Orthologs were grouped into functional categories and metabolic pathways using the BRITE hierarchy. Functional profiles were visualized in R v4.4.1 [34] using the *ggplot2* v3.5.1 [38] package. Categories included metabolism, genetic information processing, environmental information processing, cellular processes, and others. Within the "metabolism" category, xenobiotic biodegradation and secondary metabolite biosynthesis sub-pathways were analyzed independently to assess their relative contribution to the overall functional potential.

To identify enriched metabolic functions related to the degradation of aromatic hydrocarbons, KO enrichment analysis was performed using the enrichKEGG function of the *clusterProfiler* package v4.12.6 [56]. The analysis was based on a hypergeometric test with $p < 0.05$, and p-values were adjusted using the Benjamini–Hochberg method (False Discovery Rate, FDR < 0.05) to correct for multiple comparisons. KEGG pathway maps were reconstructed using KEGG Mapper to visualize the complete degradation routes of naphthalene, toluene, xylenes, benzoate, and catechol.

Additionally, a comparative mapping of KOs across samples was performed to identify sample-specific degradation potential. Only pathways with ≥70% of essential KOs detected and enzymatic steps consistently associated with taxonomically validated MAGs were considered functionally complete.

## Results

### Environmental and air quality conditions

Climatic and air quality parameters varied between sampling periods and urban sites, reflecting distinct environmental pressures on *M. indica* and its associated phyllosphere microbiota. From December 2023 (S1) to April 2024 (S2), both sites experienced higher average temperatures (from 22.96–24.28 °C in S1 to 23.86–25.11 °C in S2) and lower relative humidity and precipitation (Table 1), These changes coincided with the first major air pollution episode of 2024 in the Aburrá Valley (5).

A substantial increase in particulate matter concentrations ($PM_{2.5}$ and $PM_{10}$) was recorded in S2 compared to S1, with consistently higher values in the DT than in the SS. In S2, daily maxima of $PM_{2.5}$ reached 51.91 µg/m³ in DT and 39.94 µg/m³ in SS, while $PM_{10}$ levels reached 76.66 µg/m³ in DT. These concentrations exceed the 24-hour limits established by Resolution 2254 of 2017 (37 µg/m³ for $PM_{2.5}$ and 75 µg/m³ for $PM_{10}$), as well as the World Health Organization's 24-hour guideline values (15 µg/m³ for $PM_{2.5}$ and 45 µg/m³ for $PM_{10}$) (1), suggesting an acute environmental stressor for phyllosphere microbial communities.

Although data for $NO_2$, $O_3$, and CO were incomplete or missing from several SIATA stations during the sampling windows, available measurements indicated an overall increase in gaseous pollutants during April, in line with pollution alerts issued during that period. In particular, $NO_2$ concentrations peaked at 58.01 µg/m³ in SS during S2, alongside detectable ozone ($O_3$) and elevated carbon monoxide (CO) levels at the SS. These patterns suggest site-specific pollutant profiles that may differentially shape microbial community composition and functional potential.

BTEX concentrations, based on SIATA's passive sampling campaign in 2022, exhibited clear spatial and temporal variation. The DT consistently showed higher levels of toluene, ethylbenzene, and xylene isomers in both months, while benzene concentrations frequently exceeded the Colombian annual limit of 5 µg/m³ (Resolution 2254 of 2017). Although no specific national regulations exist for ethylbenzene and xylenes, observed levels remained below international thresholds (e.g., WHO guideline values: 22,000 µg/m³ for ethylbenzene, 870 µg/m³ for xylenes), Nonetheless, patterns suggest episodic accumulation of these compounds following pollution events (Table 2). Collectively, these findings confirm that the DT experiences more intense and variable air pollution than the SS, providing a relevant environmental framework to investigate how pollutant loads influence the taxonomic and functional structure of phyllosphere bacterial communities.

### Taxonomic profiling of phyllosphere bacteria

Amplicon sequencing of the 16S rRNA gene generated a total of 11,508,610 raw reads across all samples. After quality control and removal of non-bacterial sequences, 618,339 high-quality paired reads were retained, ranging from

**Table 1. Climatic conditions and criteria air pollutant concentrations at two urban sites in Medellín during the two sampling periods (December 2023 – S1; April 2024 – S2).**

| Sampling period – urban sites | Climatic conditions | | | Criteria air pollutants | | | | |
|---|---|---|---|---|---|---|---|---|
| | Temperature (ºC) | Relative humidity (%) | Precipitation (mm) | $PM_{2.5}$ (µg/m³) | $PM_{10}$ (µg/m³) | $NO_2$ (µg/m³) | $O_3$ (µg/m³) | CO (µg/m³) |
| S1-DT | 22.96 | 68.19 | 2.676 | 19.10 | 44.75 | NA | NA | NA |
| S1-SS | 24.28 | 68.44 | 1.008 | 14.54 | 30.34 | 50.90 | 23.31 | 2240.75 |
| S2-DT | 23.86 | 65.38 | 2.124 | 41.11 | 60.47 | 51.03 | NA | NA |
| S2-SS | 25.11 | 67.01 | 1.159 | 27.32 | 42.75 | 58.01 | NA | 1997.44 |

Notes: DT = Downtown (high pollution); SS = Southern site (lower pollution); $PM_{2.5}$ and $PM_{10}$ = Particulate Matter ≤2.5 µm and ≤10 µm in aerodynamic diameter (µg/m³); $NO_2$ = Nitrogen dioxide (µg/m³); $O_3$ = ozone (µg/m³); CO = carbon monoxide (µg/m³); NA = data not available from SIATA stations during the sampling window. Data from the Early Warning System of the Aburrá Valley (SIATA). Units correspond to mean monthly values during the respective sampling period, except for precipitation (total monthly).

**Table 2. BTEX concentrations (µg/m³) at two urban sites in Medellín during April and December 2022.**

| Date | Urban sites | BTEX (µg/m³) | | | | | |
|------|-------------|--------------|---------|--------------|----------|----------|----------|
| | | Benzene | Toluene | Ethylbenzene | o-xylene | m-xylene | p-xylene |
| Apr 2022 | DT | 3.8 | 34.5 | 8.7 | 7.7 | 12.4 | 10.3 |
| Apr 2022 | SS | 5.6 | 29.1 | 5.6 | 5.8 | 7.6 | 6.1 |
| Dec 2022 | DT | 6.8 | 22.8 | 5.7 | 5.4 | 8.7 | 7.4 |
| Dec 2022 | SS | 8.7 | 2.5 | 5.5 | 6.1 | 7.4 | 6.3 |

Notes: Data from the Early Warning System of the Aburrá Valley (SIATA), passive sampling campaign 2022. BTEX = Benzene, Toluene, Ethylbenzene, and Xylenes (o, m, p isomers) (µg/m³). Colombian anual limit for benzene = 5 µg/m³ (Resolution 2254 of 2017). WHO guideline values: 22,000 µg/m³ for ethylbenzene; 870 µg/m³ for xylenes.

31,985–76,092 reads per sample (9.1% –13.0% of raw data) (S1 Table in S1 File). Rarefaction analysis confirmed adequate sequencing depth for capturing community diversity (S2 Fig in S1 File). A total of 103,598 amplicon sequence variants (ASVs) were identified, corresponding to 38 phyla, 117 classes, 304 orders, 526 families, 1,291 genera, and 439 bacterial species, underscoring the high taxonomic diversity of the *M. indica* phyllosphere.

At the phylum level, the community was dominated by Pseudomonadota (formerly Proteobacteria) (45.9%), followed by Actinomycetota (16.4%), Bacteroidota (11.4%), Cyanobacteria (7.2%), and Deinococcota (6.1%) (Fig 1a). Within Pseudomonadota, Alphaproteobacteria (34.8%) and Gammaproteobacteria (11.1%) were the most abundant classes (Fig 1b), both of which include genera known for their roles in organic pollutant degradation [57]. Actinomycetota was dominated by the class Actinomycetia (15.2%), while Bacteroidia, Cyanobacteriia, and Deinococci were the main representatives of the remaining phyla. These dominant taxa are consistent with phyllosphere communities reported in other urban tree species [24,58–62], suggesting a degree of functional redundancy across hosts.

At the genus level, members of *Sphingomonas*, *Acidiphilium*, and *Methylobacterium* (Alphaproteobacteria); *Erwinia* and *Pantoea* (Gammaproteobacteria); *Spirosoma* and *Hymenobacter* (Bacteroidia); and *Deinococcus* (Deinococci) were particularly abundant (Fig 1e–f). Within these genera, species-level assignments such as *Sphingomonas* sp., *Massilia* sp., and *Pantoea agglomerans* were identified, Several of these taxa are recurrently reported in urban phyllosphere microbiomes and are known for their tolerance to desiccation, UV radiation, and oxidative stress, as well as for possessing metabolic pathways involved in the degradation of aromatic hydrocarbons and other xenobiotics [62–64]. Notably, sequences annotated as a *Flavobacteriales* endosymbiont were detected across multiple samples (Fig 1f), suggesting that a fraction of the phyllosphere-associated bacterial community may be transiently introduced by insects, which are frequent visitors of leaf surfaces.

Alpha diversity analyses revealed that bacterial richness was significantly higher in the DT than in the SS, based on the Shannon (p = 0.0238) and Chao1 (p = 0.0101) indices, while Simpson's index showed a similar trend (p = 0.0869) (Fig 2). In contrast, no significant differences in alpha diversity were observed between the two sampling periods (Shannon p = 0.361; Chao1 p = 0.852; Simpson p = 0.13), suggesting that site-specific conditions, rather than seasonality, played a stronger role in shaping phyllosphere bacterial richness.

These patterns indicate that urban pollution gradients may favor the enrichment of stress-tolerant and potentially pollutant-degrading taxa in the *M. indica* phyllosphere. Consequently, we evaluated whether these differences in community richness were accompanied by shifts in overall bacterial community composition through beta diversity analyses.

Beta diversity, assessed by Principal Coordinates Analysis (PcoA) of Bray–Curtis distances, revealed distinct clustering patterns of microbial communities, with significant differences between sampling periods S1 and S2 (PERMANOVA, p = 0.005, R2 = 0.7403). This effect was not driven by differences in dispersion, as indicated by a non-significant PERMDISP result (p = 0.233). A smaller but significant separation was observed between urban sites

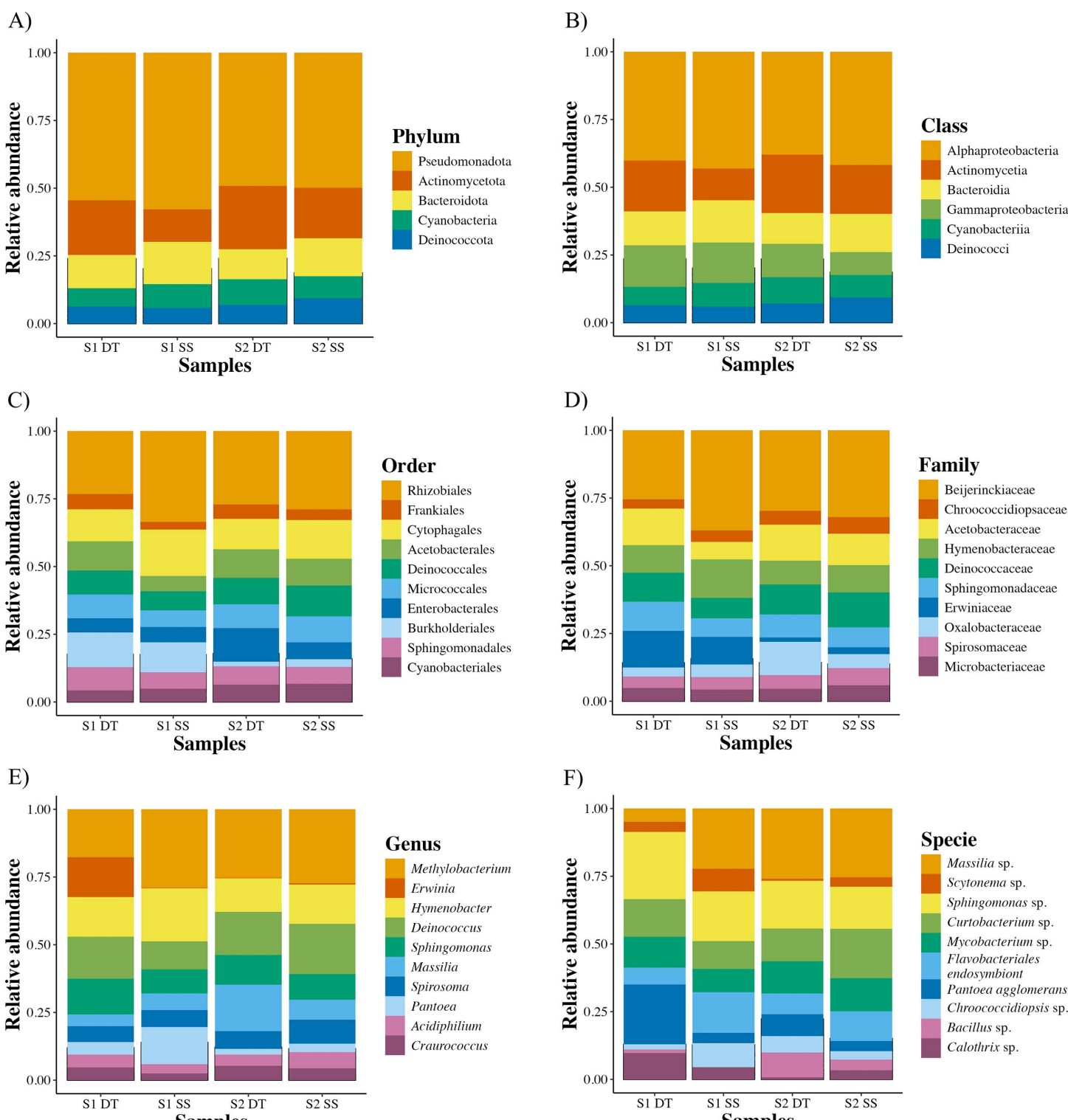

**Fig 1. Taxonomic profiling of phyllosphere bacteria.** Bar chart showing the relative abundance of the 10 most abundant ASVs at various taxonomic levels: a) phylum, b) class, c) order, d) family, e) genus, and f) species.

A)

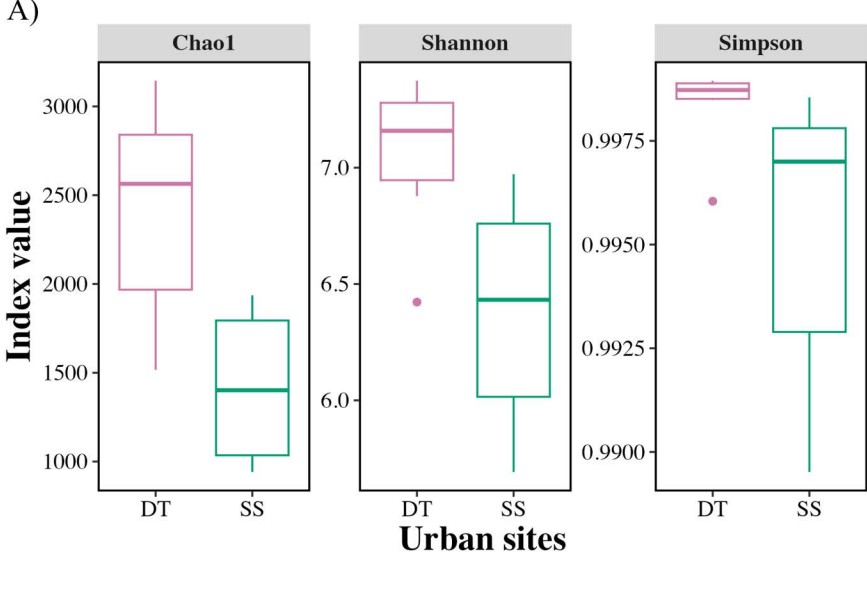

B)

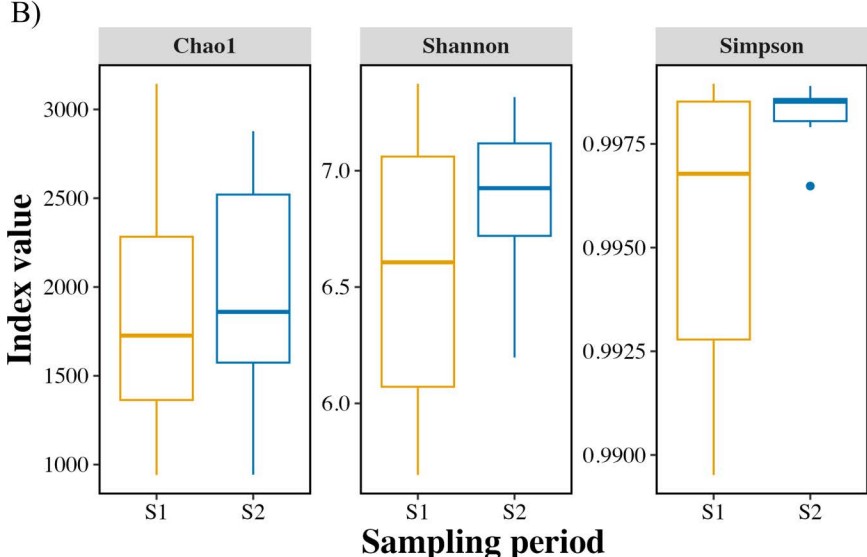

**Fig 2. Alpha diversity of the phyllospheric bacterial community.** Boxplots showing alpha diversity based on (a) urban site (downtown, DT; southern site, SS) and (b) sampling period (December 2023, S1; April 2024, S2). Diversity was assessed using Shannon, Simpson, and Chao1 indices calculated from rarefied amplicon sequence variant (ASV) counts (31,985 reads per sample). Boxplots display the median, interquartile range, and 1.5×whiskers; points represent individual samples. P-values were obtained using ANOVA.

(PERMANOVA, p = 0.036, R2 = 0.5082); however, this contrast showed heterogeneous dispersion (PERMDISP, p = 0.016) (Fig 3). Axis 1 of the PcoA explained 19.82% of the variation, primarily associated with sampling period, while Axis 2 explained 13.71%, associated with site-specific differences. These results indicate that, although spatial differences in microbial composition exist, temporal variation, likely reflecting seasonal dynamics and differential pollutant exposure, had a more pronounced effect on community structure.

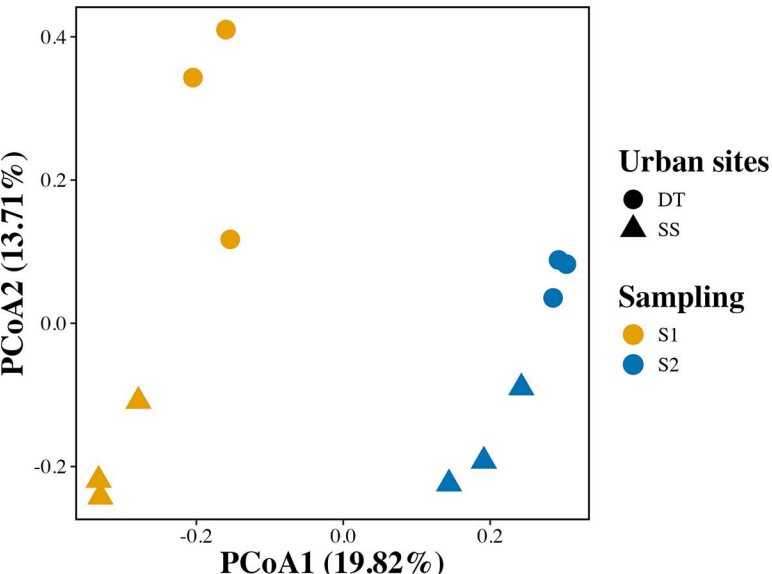

**Fig 3. Beta diversity of the phyllosphere bacterial community in *Mangifera indica*.** Principal Coordinates Analysis (PcoA) based on Bray–Curtis dissimilarities of amplicon sequence variant (ASV) relative abundances. Samples were collected from the downtown (DT) and southern site (SS) during December 2023 (S1) and April 2024 (S2). Axis 1 and Axis 2 explain 19.82% and 13.71% of the variation, respectively. Group differences were assessed using PERMANOVA (999 permutations), with significant separation by sampling period (p = 0.005) and site (p = 0.036).

## Metagenomic sequencing output and quality filtering results

Shotgun metagenomic sequencing yielded a total of 497,578,986 raw reads across all samples, with individual libraries ranging from 33.2 to 51.0 million reads. After quality control and removal of host-associated sequences, 460,284,726 high-quality reads were retained, corresponding to 92.5% of the initial dataset— reflecting both efficient DNA recovery and sequencing performance (S2 Table in S1 File).

Assembly of the filtered reads using metaSPAdes produced between 9,280 and 394,109 scaffolds per sample (S3 Table in S1 File). This variability likely reflects differences in microbial community complexity, genome size heterogeneity, and sequencing depth. Binning and refinement steps generated between 2 and 10 metagenome-assembled genomes (MAGs) per sample. After applying quality filters (≥50% completeness, <10% contamination), a total of 54 MAGs were recovered across all datasets. Of these, 10 MAGs met the criteria for high-quality genomes (≥90% completeness, <5% contamination) according with MIMAG standards [65] (S4 Table in S1 File).

The recovery of high-quality MAGs from both urban sites demonstrates that the phyllosphere of *M. indica* harbors sufficiently abundant and diverse bacterial genomes suitable for downstream functional profiling (S5 Table in S1 File). Although the number of MAGs varied between samples, no clear spatial or temporal clustering was observed at the assembly level, suggesting that functional gene repertoires (section 3.4) may capture ecologically relevant differences more effectively than contig-level metrics alone. These MAGs provided the basis for functional annotation and pathway reconstruction, enabling the identification of microbial genes associated with the degradation of airborne aromatic pollutants.

## Functional profiling of the metagenome-assembled genomes (MAGs)

**General function profiles.** Functional annotation of recovered (MAGs) revealed a predominance of genes associated with core bacterial processes. Across all samples, metabolic functions represented the largest category (54.4% – 65.5%),

followed by genetic information processing (11.4% – 19.5%), environmental information processing (13.4% – 20.1%), and cellular processes (5.1% – 7.4%) (Fig 4a). Functions related to organismal systems and human diseases were minimally represented (<2%).

Within the metabolic category, carbohydrate metabolism (28.1% – 29.4%) was most abundant, followed by amino acid metabolism (12.9% – 16.7%), cofactor and vitamin biosynthesis (12.4% – 15.0%), energy metabolism (10.3% – 11.8%),

A)

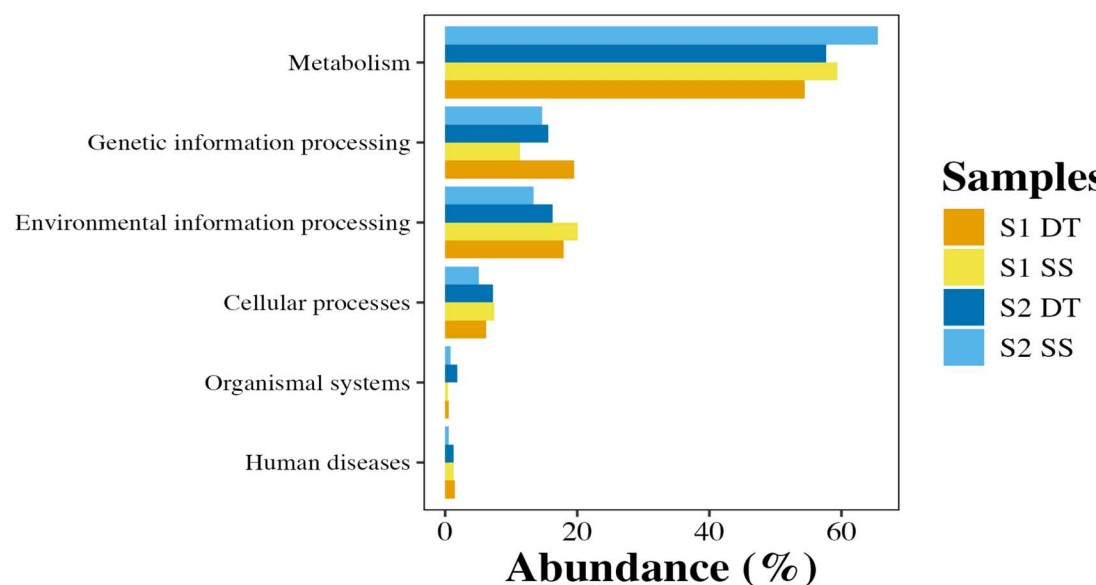

B)

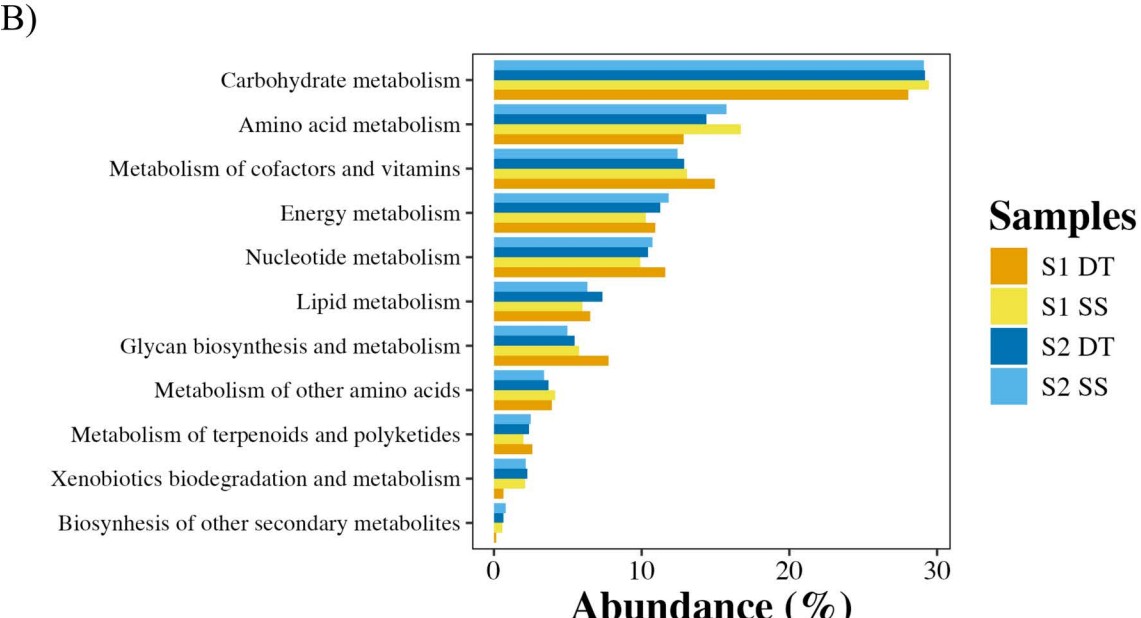

**Fig 4. General functional profiles of MAGs.** Bar chart showing the abundance of a) major functional pathways and b) metabolic-related functional categories.

and nucleotide metabolism (9.9% – 11.6%). Genes linked to lipid metabolism, glycan biosynthesis, and the biosynthesis of secondary metabolites were less represented. Pathways related to xenobiotic biodegradation accounted for 0.6% – 2.3% of the annotated KEGG Orthologs (KOs), supporting the inference that phyllosphere bacteria possess genetic features potentially associated with adaptation to polluted urban environments (Fig 4b). These functional patterns were consistent across samples, although a slightly higher representation of xenobiotic-related functions was detected in the SS, suggesting a possible enrichment in genetically inferred pollutant degradation potential.

**Aromatic compound degradation pathways.** KEGG KO enrichment analysis revealed statistically significant overrepresentation (FDR < 0.05) of genes involved in the degradation of aromatic hydrocarbons, including naphthalene, toluene, xylenes, benzoate, and catechol, particularly in samples from the SS (S3 Fig in S1 File).

In these samples, MAGs affiliated with *Methylobacterium longum* carried the *nahAa* gene (K14581), encoding the ferredoxin reductase subunit of naphthalene 1,2-dioxygenase. This enzyme initiates the oxidation of naphthalene to salicylate, a key step in aromatic ring cleavage. For toluene and xylenes, the presence of *xylC* gene (K00141), encoding benzaldehyde dehydrogenase, was detected in *Pseudomonas_E palleroniana*, indicating a genetically inferred capacity to participate in the transformation of methylated benzaldehydes to benzoate. Further downstream, *Serratia ficaria* MAGs carried the *benA-xylX*, *benB-xylY*, and *benC-xylZ* (K00055, K05550, K05784) genes encoding the multicomponent benzoate dioxygenase system, as well as *benD-xylL* (K05783), suggesting the genetic potential to convert benzoate into catechol.

Catechol degradation was genetically represented via two complementary pathways (Fig 5). In samples from S1 (south site), the ortho-intradiol cleavage pathway was dominant, mediated by genes *catA*, *catB*, and *catC*, found in MAGs of *Serratia*, *Pseudomonas*, and *Achromobacter* (Pseudomonadota) (S6 Table in S1 File). In contrast, the meta-extradiol pathway, including genes *catE*, *dmpB*, *dmpC*, and *mhpD*-F, was identified primarily in S2 MAGs affiliated with *Actinomycetospora* and *Nakamurella* (Actinomycetota), suggesting potential functional differentiation between sampling periods (S7 Table in S1 File).

Pathway completeness analysis revealed that in SS, over 80% of essential KOs required for benzoate and catechol degradation were present, indicating the potential for near-complete mineralization of these compounds. In contrast, DT samples exhibited only partial presence of these pathways, with fewer complete KO sets and lower MAG association. Based on DNA-level analyses, our results indicate that the genetic repertoire required for complete catechol degradation is fully present in communities from less contaminated sites, whereas only partial pathway reconstruction was possible in communities from more polluted environments. Similar patterns have been reported in other low-biomass surface-associated microbiomes, where chronic environmental stress is associated with a loss or fragmentation of metabolically costly pathways [66].

**Integrative patterns across environmental, taxonomic, and functional datasets.** Taken together, our results demonstrate that *M. indica* phyllosphere communities in Medellín are shaped by the interplay between site-specific air pollution levels and temporal environmental variation. Elevated particulate matter and BTEX concentrations during high-pollution episodes coincided with measurable shifts in bacterial richness, community composition, and functional potential. Genera with documented stress tolerance and xenobiotic degradation capacity were consistently detected, yet the completeness of aromatic hydrocarbon degradation pathways was greater in the less polluted SS. This pattern suggests that chronic exposure to elevated pollutant loads in the DT may favor stress-adapted specialists over metabolically versatile degraders [66]. The stress experienced by phyllosphere-associated bacteria is not driven solely by exposure to high concentrations of atmospheric organic pollutants, but also by additional environmental pressures inherent to the phyllosphere. Such contrasts underscore the ecological sensitivity of phyllosphere microbiota to urban air quality and their relevance as both bioindicators and reservoirs of biotechnologically valuable functions. These findings provide a framework to interpret the ecological mechanisms discussed below and to explore the potential of phyllosphere bacteria as nature-based solutions for air pollution mitigation in tropical cities.

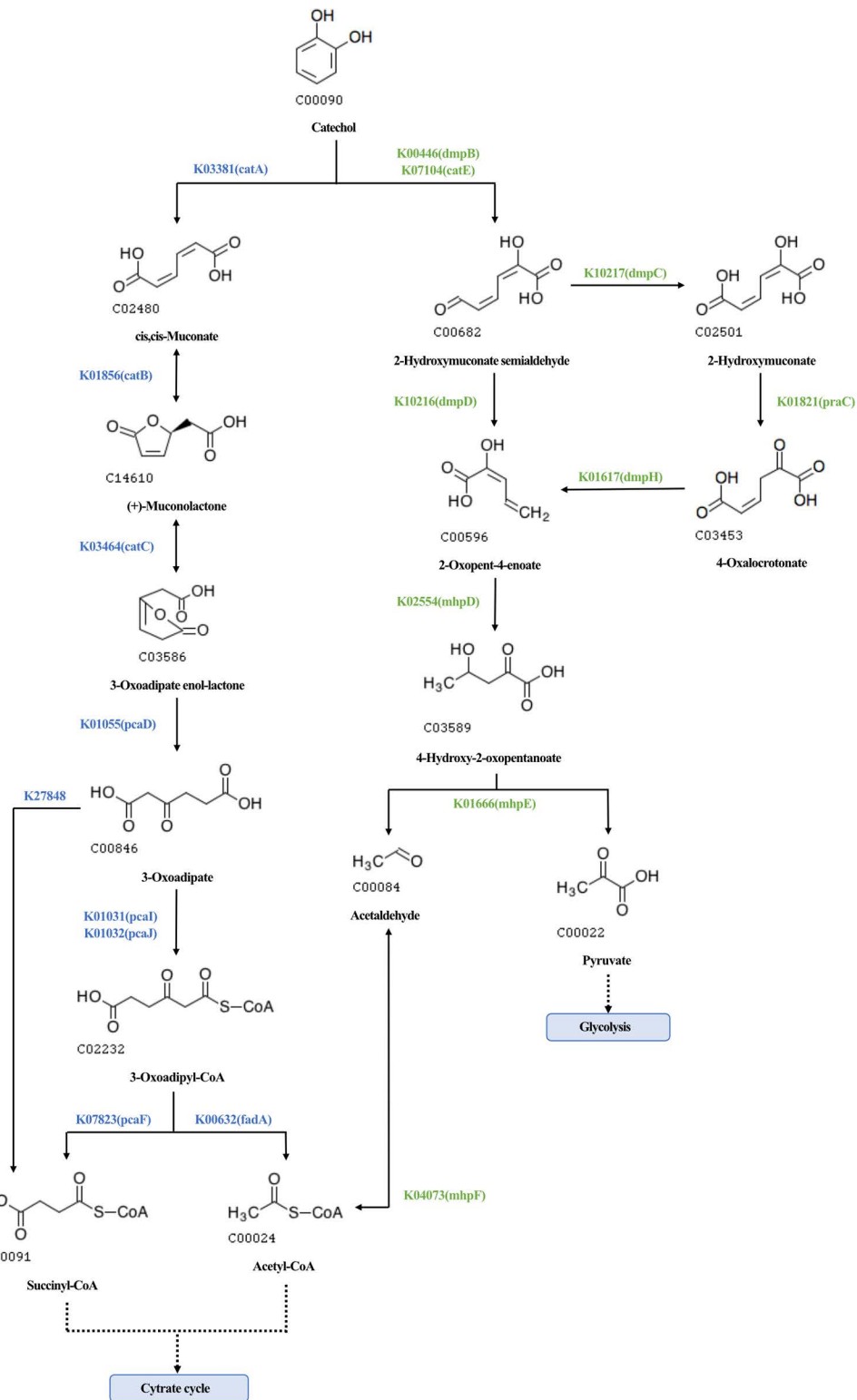

**Fig 5. Catechol degradation pathways identified in *Mangifera indica* phyllosphere MAGs.** Reconstruction of catechol degradation routes based on KEGG Mapper, highlighting the ortho-intradiol cleavage pathway (blue) and the meta-extradiol cleavage pathway (green). Enzymes are labeled with their KEGG Orthology (KO) identifiers. Pathways were inferred from metagenome-assembled genomes (MAGs) meeting the quality thresholds of ≥50% completeness and <10% contamination, with taxonomic affiliations indicated in the text.

## Discussion

This study addressed the taxonomic and functional composition of *M. indica* phyllosphere bacterial communities in two urban sites of Medellín with contrasting air pollution levels, integrating amplicon and metagenomic sequencing approaches. By linking environmental parameters, taxonomic profiles, and functional gene repertoires, we demonstrated that seasonal pollution episodes exert a stronger influence on microbial community structure than spatial differences, while also shaping the functional potential for aromatic hydrocarbon degradation. These findings expand the current understanding of tropical urban phyllosphere microbiomes, which remain underrepresented in global datasets.

The Aburrá Valley represents a complex atmospheric system, where the interaction between topography, meteorology, and anthropogenic emissions results in recurrent pollution episodes. These episodes, typically occurring between February and March, coincide with the seasonal transition from dry to rainy periods, which restricts pollutant dispersion and promotes accumulation of fine particulate matter ($PM_{2.5}$, $PM_{10}$) and volatile organic compounds (BTEX) [5,6]. Similar trends have been observed in other cities worldwide, where meteorological stability and limited ventilation enhance air pollution, such as haze formation events in China [67] and $PM_{10}$ accumulation in Europe during cold seasons [16].

The pollution peaks recorded in April 2024 were strongly associated with shifts in microbial community composition, supporting previous findings that phyllosphere microbiota are sensitive to air quality fluctuations [19,21]. The concurrent increase in gaseous pollutants ($NO_2$, $O_3$, CO) suggests a multifactorial stress scenario, where oxidative and osmotic pressures shape microbial succession by favoring stress-tolerant and metabolically versatile taxa.

The predominance of Pseudomonadota and Actinomycetota across all samples aligns with findings from multiple phyllosphere studies in temperate [28,68,69], subtropical [70], and neotropical ecosystems [71]. Within Pseudomonadota, Alphaproteobacteria and Gammaproteobacteria dominated, reflecting their ability to degrade aromatic hydrocarbons such as BTEX and PAHs [57]. Actinomycetia, the dominant class within Actinomycetota, is recognized for biosurfactant secretion that increases hydrocarbon bioavailability [72,73]. These bacterial groups have also been reported in the phyllosphere of various urban trees, including *Platanus × acerifolia* [20,74], *Acer pseudoplatanus* (20), and *Carpinus betulus* (18), suggesting that their ecological role in colonizing aerial plant surfaces is widespread across ecosystems and plant hosts.

Alpha diversity was higher in DT samples, potentially reflecting chronic exposure to complex pollutant mixtures that promote niche differentiation and microbial coexistence. Beta diversity, however, showed stronger temporal than spatial differences, in agreement with previous studies reporting that seasonal variation exerts a greater effect on phyllosphere communities than geographic location [16,75]. Nonetheless, other works suggest that rural–urban contrasts and local pollution intensity can also be key determinants of microbial assemblages [19–21,59].

Metagenomic analyses revealed diverse metabolic functions, with carbohydrate and amino acid metabolism being central to bacterial survival on leaf surfaces [14,76]. Importantly, we detected key genes involved in the degradation of aromatic hydrocarbons, including *nahAa*, *xylC*, *benA*, *benD*, *catA*, and *catE*. These genes participate in the transformation of compounds such as naphthalene, toluene, xylenes, and benzoate into catechol, which can subsequently be metabolized through different ring-cleavage pathways [77,78].

The detection of both ortho-intradiol (C12D) and meta-extradiol (C23D) cleavage pathways demonstrates functional redundancy—a well-known microbial strategy to maintain ecosystem resilience under fluctuating stress [79]. The ortho pathway, dominant in the first sampling period, was mainly associated with *Pseudomonas*, *Serratia*, and *Achromobacter*, while the meta pathway appeared later in Actinomycetota, suggesting functional succession driven by changing environmental conditions.

Specific taxa carried genes of relevance for bioremediation. For instance, *Methylobacterium longum* harbored *nahAa*, a component of the naphthalene dioxygenase system, consistent with reports of this genus metabolizing VOCs and colonizing leaf surfaces [80–82]. Similarly, *Pseudomonas*_E *palleroniana* encoded *xylC*, involved in xylene degradation [22], and *Serratia ficaria* carried genes for benzoate and catechol metabolism, as previously documented in petroleum-contaminated environments [17,18].

The overlap between taxonomic dominance and functional potential in *Methylobacterium*, *Pseudomonas*, and *Serratia* reinforces their ecological role in atmospheric remediation. These genera have repeatedly been associated with VOC degradation, high metabolic plasticity, and colonization of urban trees [16,18]. By integrating metagenome-assembled genome (MAG) annotations with taxonomic assignments, this study reduced the uncertainty typical of amplicon-based analyses and provided stronger ecological interpretations of biodegradative capacity in the phyllosphere of *M. indica*.

Beyond their ecological significance, phyllosphere microbiota constitute a valuable reservoir of biodegradative genes with promising applications in biotechnology. The recovery of complete or near-complete aromatic hydrocarbon degradation pathways in MAGs highlights their potential for biosensor development, microbial bioaugmentation, or genetic prospecting for environmental engineering.

Our findings support the idea of enhancing green infrastructure with plant–microbe consortia as part of nature-based strategies (NbS) for urban air quality improvement [19,79]. However, the feasibility of such interventions in tropical cities requires further assessment of host–microbe stability, long-term persistence, and ecosystem-level impacts.

## Conclusion

This study provides the first integrated taxonomic and functional characterization of *Mangifera indica* phyllosphere bacteria along urban air pollution gradients in Medellín, Colombia. Combining 16S rRNA amplicon sequencing and shotgun metagenomics, we show that these communities are taxonomically diverse and harbor genetic potential for xenobiotic degradation, as evidence by the presence of genes and complete metabolic pathways associated with the transformation of airborne pollutants, including naphthalene, toluene, xylenes, and benzoate. Functional diversity was shaped by both spatial and temporal factors, with seasonal pollution episodes exerting stronger effects on community composition than site location.

The detection of genes associated with both ortho- and meta-cleavage catechol pathways highlights their potential metabolic plasticity and resilience under fluctuating urban pollution conditions. While our findings do not demonstrate in situ pollutant degradation, they position urban tree–associated microbiomes as promising reservoirs of genetic potential relevant to nature-based strategies for atmospheric bioremediation and sustainable air quality management in tropical cities. Future studies integrating transcriptomic approaches or enzyme-level assays will be essential to validate pathway expression and confirm functional activity. These insights also open opportunities to explore the contribution of diverse tree species and rare microbial taxa, and to test the applicability of this approach in other tropical urban centers.

## Supporting information

**S1 File. S1 Fig. Map of Medellín showing the distribution of *Mangifera indica* in urban sites (downtown, DT; southern site, SS), along with meteorological stations (blue) and air quality stations (orange) operated by SIATA.** S2 Fig. Rarefaction curves showing observed richness across samples at different sequencing depths. S3 Fig. Dot plot illustrating enriched aromatic compound degradation pathways in urban sites: a) DT S1, b) DT S2, c) SS S1, and d) SS S2. S1 Table. Summary of 16S rRNA gene amplicon sequencing output and quality-control processing statistics. S2 Table. Summary of shotgun metagenomics sequencing output and quality-control metrics. S3 Table. Assembly performance metrics derived from high-quality reads merged into scaffolds. S4 Table. Percentage of bin completeness and contamination in the samples (sampling period – urban sites) after bin refinement. S5 Table. Gene prediction outcomes generated using PROKKA and associated annotation statistics. S6 Table. Genes identified at S1 SS involved in catechol degradation via the ortho-intradiol cleavage pathway. S7 Table. Genes identified at S1 SS involved in catechol degradation via the meta-extradiol cleavage pathway.
(ZIP)

## Acknowledgments

We thank the Early Warning System of the Aburrá Valley (SIATA) for providing air quality data for Medellín, and the G-LIMA research group at University of Antioquia for their support during sampling activities.

## Author contributions

**Conceptualization:** Natalia Bernal Hernández, Nancy J. Pino, Luisa María Múnera Porras.

**Data curation:** Natalia Bernal Hernández, Héctor Alejandro Rodríguez Cabal, Sara Ramírez Restrepo.

**Formal analysis:** Natalia Bernal Hernández, Héctor Alejandro Rodríguez Cabal, Nancy J. Pino, Sara Ramírez Restrepo, Luisa María Múnera Porras.

**Funding acquisition:** Nancy J. Pino, Luisa María Múnera Porras.

**Investigation:** Natalia Bernal Hernández, Nancy J. Pino, Luisa María Múnera Porras.

**Methodology:** Natalia Bernal Hernández, Héctor Alejandro Rodríguez Cabal, Sara Ramírez Restrepo, Luisa María Múnera Porras.

**Project administration:** Luisa María Múnera Porras.

**Resources:** Nancy J. Pino, Luisa María Múnera Porras.

**Software:** Héctor Alejandro Rodríguez Cabal.

**Supervision:** Héctor Alejandro Rodríguez Cabal, Nancy J. Pino, Sara Ramírez Restrepo, Luisa María Múnera Porras.

**Validation:** Héctor Alejandro Rodríguez Cabal, Sara Ramírez Restrepo.

**Writing – original draft:** Natalia Bernal Hernández.

**Writing – review & editing:** Héctor Alejandro Rodríguez Cabal, Nancy J. Pino, Sara Ramírez Restrepo, Luisa María Múnera Porras.

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
