## [Decision Letter · Decision Letter 0]

29 Oct 2025

*Mangifera indica* in response to urban air pollutants in Medellín, Colombia.in response to urban air pollutants in Medellín, Colombia.PLOS ONE

Dear Dr. Múnera,

Thank you for submitting your manuscript to PLOS ONE. After careful consideration, we feel that it has merit but does not fully meet PLOS ONE’s publication criteria as it currently stands. Therefore, we invite you to submit a revised version of the manuscript that addresses the points raised during the review process.

We look forward to receiving your revised manuscript.

Kind regards,

Massimiliano Cardinale, PhD

Academic Editor

PLOS ONE

Journal Requirements:

1. Please ensure that your manuscript meets PLOS ONE’s style requirements, including those for file naming. The PLOS ONE style templates can be found at

https://journals.plos.org/plosone/s/file?id=wjVg/PLOSOne_formatting_sample_main_body.pdf andandandand

https://journals.plos.org/plosone/s/file?id=ba62/PLOSOne_formatting_sample_title_authors_affiliations.pdf....

4. We note that Figures S1 in your submission contain [map/satellite] images which may be copyrighted. All PLOS content is published under the Creative Commons Attribution License (CC BY 4.0), which means that the manuscript, images, and Supporting Information files will be freely available online, and any third party is permitted to access, download, copy, distribute, and use these materials in any way, even commercially, with proper attribution. For these reasons, we cannot publish previously copyrighted maps or satellite images created using proprietary data, such as Google software (Google Maps, Street View, and Earth). For more information, see our copyright guidelines: http://journals.plos.org/plosone/s/licenses-and-copyright.

a. You may seek permission from the original copyright holder of Figures S1 to publish the content specifically under the CC BY 4.0 license.

Reviewers' comments:

Reviewer's Responses to Questions

**Comments to the Author**

1. Is the manuscript technically sound, and do the data support the conclusions?

Reviewer #1: Yes

Reviewer #2: Partly

Reviewer #3: Yes

Reviewer #4: Yes

2. Has the statistical analysis been performed appropriately and rigorously?

Reviewer #1: Yes

Reviewer #2: Yes

Reviewer #3: Yes

Reviewer #4: Yes

3. Have the authors made all data underlying the findings in their manuscript fully available?

Reviewer #1: Yes

Reviewer #2: No

Reviewer #3: Yes

Reviewer #4: Yes

4. Is the manuscript presented in an intelligible fashion and written in standard English?

Reviewer #1: Yes

Reviewer #2: Yes

Reviewer #3: Yes

Reviewer #4: Yes

Reviewer #1: I commend the authors for conducting a novel and technically ambitious study exploring how urban air pollution influences the phyllosphere microbiome of Mangifera indica. The integration of both 16S rRNA amplicon and shotgun metagenomic sequencing provides valuable ecological and functional insights. The manuscript is generally well structured, and the data are potentially of broad relevance to microbial ecology, environmental health, and urban sustainability research.

That said, several methodological and interpretative aspects require clarification or expansion to strengthen the robustness and reproducibility of the findings. Specifically, greater detail is needed on the sampling design, contamination controls, MAG quality, and statistical reporting. Additionally, the language can be refined for conciseness and clarity, and the conclusions should more cautiously reflect the evidence presented.

Addressing these issues through revision will substantially improve the scientific soundness and readability of the paper. I appreciate the authors’ efforts and hope these comments are helpful toward enhancing the manuscript’s quality.

Major Comments

Sampling design and replication:

The manuscript should explicitly state whether the triplicate samples collected per site and time represent true biological replicates (distinct trees) or technical replicates from the same individual. This distinction is critical for assessing statistical independence and ecological inference.

Contamination controls in phyllosphere samples:

Low-biomass leaf-surface samples are prone to reagent contamination. Although negative controls are mentioned, the authors should present a short description or table showing the taxa detected in blanks, the number of ASVs removed, and evidence that blanks had negligible impact on the final community composition.

MAG quality and pathway inference:

For metagenome-assembled genomes (MAGs) cited as carrying aromatic hydrocarbon degradation genes, provide completeness, contamination, and coverage values. Claims about “complete pathways” should rely only on MAGs with ≥70–80% completeness and <10% contamination. A supplementary table summarizing MAG metrics and associated KOs would improve transparency.

Functional inference limitations:

The conclusion that the phyllosphere “harbors robust genetic potential for xenobiotic degradation” should be moderated. The presence of catabolic genes indicates potential rather than confirmed activity. The authors should explicitly acknowledge this limitation and suggest follow-up functional validation (e.g., transcriptomics or enzyme assays).

Statistical transparency:

The diversity analyses (Shannon, Chao1, PERMANOVA) are appropriate but lack reporting of parameters such as the number of permutations, distance metric, and tests for dispersion (PERMDISP). Reporting these values and including R² effect sizes will enhance interpretability.

BTEX and pollution data alignment:

BTEX data were derived from 2022 monitoring, whereas sampling occurred in late 2023 and early 2024. The authors should clarify this temporal mismatch and discuss how it might influence interpretations linking pollutant levels to microbial gene content.

Reproducibility and supplementary materials:

Include a detailed supplementary dataset showing per-sample sequencing metrics (raw reads, host-filtered reads, retained reads, number of scaffolds, MAG counts) and key gene/pathway completeness scores. This will ensure full reproducibility of the analyses.

Minor Comments

Abstract:

Indicate the number of samples (N = 12) and briefly mention sequencing platforms (MiSeq for 16S, NovaSeq for shotgun).

Phrase functional conclusions more cautiously, emphasizing “genetic potential.”

Language and presentation:

The manuscript is well organized but needs minor English editing for grammar, tense consistency, and readability. Simplify complex sentences in the Abstract and Discussion and ensure consistent use of abbreviations (DT, SS, S1, S2).

Figures and tables:

Expand figure legends to define all abbreviations, sample sizes, and units.

Ensure table units (e.g., CO concentrations) are clearly stated and consistent.

Increase font size for axis labels and use colorblind-friendly palettes.

Taxonomic naming:

Consider noting Pseudomonadota (Proteobacteria) at first mention for clarity to readers familiar with classical taxonomy.

Ethical/collection statement:

Add a brief note confirming institutional or municipal permission for sampling plant material in Medellín.

Data availability statement:

The accession number (PRJNA1269120) should be cross-checked for functionality and included in the Data Availability section as a live link.

Reviewer #2: The authors analyzed the bacterial phyllosphere community of Mangifera indica trees at two different sites, one with better air quality and one with poorer air quality. Sampling was conducted at two separate time points; however, it is unclear how many trees were sampled in total and whether the same trees were sampled at both time points. Additionally, it is not specified whether the biological replicates refer to individual trees or composite samples from a single tree. If the latter, this would imply that only two trees were sampled—one at each location (if the same trees were sampled at both sampling times). This could mean that differences observed between sites are primarily due to microbiome variation between individual trees rather than true site-related differences. However, I assume that this is not the case. Anyway, clarification of the sampling strategy is needed.

The authors report a lower abundance of xenobiotic biodegradation pathways at the more polluted sites, which appears counterintuitive. They do not comment on whether these differences are statistically significant. Currently, the only differences noted are in community structure between sites and higher alpha diversity at the more polluted sites. It would be beneficial to include analyses focusing on features that are exclusively present at the more polluted sites, maybe using a differential abundance analysis to identify potential marker taxa. Additionally, matching the MAGs with these marker taxa could facilitate the investigation of specific strains for interesting pathways and metabolic capabilities. Such findings could significantly enhance the study’s value.

While the metagenomic data have been deposited in the NCBI Sequence Read Archive, the raw amplicon data have not been made publicly available? Clarification is needed, otherwise the statement indicating that all data are freely accessible is inaccurate.

The manuscript's writing style is clear; however, the figures could be improved with more intuitive color schemes to facilitate interpretation. The paper would benefit from a better integration of the metagenomic and amplicon datasets to find novel results. Once all issue and comments (find more comments below) are resolved, I believe the paper is suitable for publication in PLOS ONE.

Comments:

Lines 180-182: If the negative controls are only used to check the dataset for ASVs exclusively present in the negative controls, then no contaminants are removed from the dataset at all. Maybe the negative controls can be shown on a PCOA plot together with the other samples. If they do not cluster with the other samples, then there should be no problem.

Lines 118-136: I am not sure if I understood the sampling strategy correctly. More details would be useful. For example, it is written that 12 samples were taken. Does this refer to 12 different trees? Were the same trees sampled again at S2 or different trees? Is one biological replicate one tree? Or were multiple composite sample take from one tree?

Lines 325-326: “both of which include genera known for their roles in organic pollutant degradation.” This statement needs a reference.

Lines 337-338: The authors refer here to Fig 1e-f and 1f is showing the species, but the text is only focusing on the genera, which is a bit misleading. I would suggest to either remove species (1f) from the figure or also mention the species in the text. It would anyway be interesting to talk about the species shown in 1f since there is also the Flavobacteriales endosymbiont, indicating that a significant proportion of the community are insect endosymbionts? I think this is a point for the discussion.

Line 331: Figure 1: In general, I am wondering what exactly is shown in figure 1. I cannot believe that the whole bacterial community are only 10 different genera or 10 different species. In your alpha diversity plots we see an estimated chao richness of 1000 up to 3000 different ASVs…. Are only taxa shown above a certain abundance threshold? Why not summarize the rest in “Other” and give an impression about the whole bacterial community? Please revise Figure 1 or at least state in the figure legend what is shown. Otherwise, the figure is quite misleading telling us that there are only 10 different species in the phyllosphere of Mangifera indica.

Line 351: Figure 2: In my opinion, figure 2 would be easier to read if different color schemes are used depending what is compared. I would suggest to use different colors for the sites and the sampling times, and not to use the same colors for sites and sampling times.

Line 374: Figure 3: Since the two different colors are used for the sampling times, the focus of the figure is more on the sampling times and not on the sites. Not sure if this is intended. I would suggest to use the different colors to differentiate between the sites, which is the focus of the whole story.

Line 417: Figure 4: In my opinion, the coloring and the order of the bars is misleading. Again, the focus of this figure seems to be on the comparison of sampling time 1 against sampling time 2. I would suggest to change the order to DT S1, DT S2, SS S1, SS S2. If there are significant differences, I would suggest to indicate that in the figure.

Lines 423-429: I am not sure if the statements made here are correct. It is written that the presence of xenobiotics degradation pathways supports the hypothesis of phyllosphere bacteria being equipped with these pathways. Well yes, but would this not mean that in the more polluted environment, the abundance of the xenobiotics degradation pathways should be higher, not the other way around as suggested in this work? The abundance of xenobiotics degradation pathways is clearly less abundant in the more polluted DT S1 compared to the less polluted SS S1, as shown in figure 4B.

Line 455: Mangifera indica not italic

Reviewer #3: The study reported the taxonomic and functional composition of Mangifera indica, a species of phyllosphere bacteria, and their relationship with urban air pollution in an urban tropical location, Medellín, Colombia. The authors collected samples from tree leaves (n = 12) in two seasons at two urban sites to conduct an analysis of the microbial communities using multi-omics approaches (i.e., 16S rRNA amplicon sequencing and shotgun metagenomics). The authors found taxonomically diverse communities that harbor complete pathways for degrading airborne pollutants at two sites, and functional diversity was shaped by both spatial and temporal factors. Overall, the methods are validated, and the findings have implications for bioremediating urban air pollution. However, some issues need to be addressed before considering publication in PLOS One. My specific comments are as follows:

1. Line 80: Define the acronyms BTEX and PAHs at their first occurrence.

2. Line 243: Define the acronym FDR at its first occurrence.

3. Lines 254–309: Subscript the numbers for the terms of air pollutants, including PM2.5, PM10, NO2, and O3; report corresponding standard deviations for their average levels; and perform statistical difference tests for their levels between seasons and sites to support the site-specific arguments.

4. Results writing style: Present the results logically instead of in a figure-by-figure manner.

5. Line 402: Where is section 3.4?

6. Color scheme for Figure 1: Use different color schemes for different taxonomic levels.

Reviewer #4: This study investigates how urban pollution gradients influence the taxonomic and functional composition of Mangifera indica phyllosphere bacterial communities. Overall, this is a nice study and a well-organized draft, with a clear structure, detailed methodological description, and accurate presentation of results; however, a few minor issues remain that should be addressed before publication.

1.The introduction is informative and well structured, but it is overly detailed, particularly in the global and regional background sections. The extensive statistics and descriptive information make the main research question less focused. This section would benefit from condensing the general context, improving transitions between ideas, and more clearly emphasizing the specific knowledge gap and study objectives. Overall, a shorter and more targeted introduction would enhance both readability and impact.

A similar issue of excessive detail is also present in the Methods section. I recommend reviewing the paper (PMID: 35094961) as an example of how to describe analytical procedures clearly and concisely.

2. In the section “Taxonomic characterization of phyllosphere bacteria,” please clarify whether the removal of low-abundance ASVs was performed before or after rarefaction, as the current description leaves the sequence of procedures unclear.

3. For Tables 1 and 2, please consider visualizing the data as line charts, adding WHO standard reference lines to improve readability and facilitate comparison.

4. In Figure 1f, all taxa are shown at the species level and sum to 1. However, the sequencing method used in this study—the 16S rRNA V3–V4 region—generally does not provide sufficient resolution for reliable species-level identification. Please clarify how these taxa were assigned to the species level.

5. There is some redundancy in describing data processing steps in both the Methods and Results sections. Since these procedures are already detailed in the Methods, consider streamlining the Results section to focus more directly on the findings rather than methodological repetition.

.

Reviewer #1: No

Reviewer #2: No

Reviewer #3: No

Reviewer #4: No

---

## [Author Response · Author response to Decision Letter 1]

17 Feb 2026

The response to each comment or observation is in the attached file: "Response to Reviewers_PLOS ONE_14.12.2025".

---

## [Decision Letter · Decision Letter 1]

2 Mar 2026

Dear Dr. Múnera,

Thank you for submitting your manuscript to PLOS ONE. After careful consideration, we feel that it has merit but does not fully meet PLOS ONE’s publication criteria as it currently stands. Therefore, we invite you to submit a revised version of the manuscript that addresses the points raised during the review process.

We look forward to receiving your revised manuscript.

Kind regards,

Massimiliano Cardinale, PhD

Academic Editor

PLOS One

Journal Requirements:

Reviewer's Responses to Questions

**Comments to the Author**

Reviewer #1: All comments have been addressed

Reviewer #2: (No Response)

Reviewer #3: All comments have been addressed

Reviewer #4: All comments have been addressed

2. Is the manuscript technically sound, and do the data support the conclusions?

Reviewer #1: Yes

Reviewer #2: No

Reviewer #3: Yes

Reviewer #4: Yes

3. Has the statistical analysis been performed appropriately and rigorously?

Reviewer #1: Yes

Reviewer #2: No

Reviewer #3: Yes

Reviewer #4: Yes

4. Have the authors made all data underlying the findings in their manuscript fully available?

Reviewer #1: Yes

Reviewer #2: Yes

Reviewer #3: Yes

Reviewer #4: Yes

5. Is the manuscript presented in an intelligible fashion and written in standard English?

Reviewer #1: Yes

Reviewer #2: Yes

Reviewer #3: Yes

Reviewer #4: Yes

Reviewer #1: (No Response)

Reviewer #2: Dear authors,

Thank you for addressing all of my comments. Unfortunately, however, with sampling only one individual tree per site, it is impossible to say if the pollution is the driver of the observed differences in the phyllosphere microbiome, or if the differences are primarily due to microbiome variation between individual trees.

You mentioned the following publication, in which a similar sampling strategy was used according to you: Laforest-Lapointe I, Messier C, Kembel SW. Host species identity, site and time drive temperate tree phyllosphere bacterial community structure.

I am familiar with that study. In their research, Laforest-Lapointe et al. sampled three times during the 2013 growing season (June, July, and August) from three individual trees per species at each site, totaling 180 samples. They also found that individual trees harbored distinct microbial communities.

You mention in your comments to the reviewer:

Only one tree per site was sampled, and the same individuals were sampled at both time points. This design was intentionally chosen to examine within-tree temporal dynamics while minimizing confounding variation introduced by differences among host individuals—a factor known to strongly influence phyllosphere microbiome composition.

However, within-tree temporal dynamics are not shown in the manuscript. Which, again, is hard to do with only to sampling timepoints.

It may be worthwhile to examine whether individual amplicon sequence variants (ASVs) show any response to particulate matter (PM) levels, potentially through Spearman correlations or similar statistical approaches. Identifying strains that exhibit significant associations with PM values could help determine if these strains may be driving differences in the genetic repertoire related to xenobiotic degradation.

Further, you could consider splitting the dataset by individual trees to investigate whether temporal changes in the microbiome occur within each tree. Comparing these changes across trees can help determine if the microbiome’s temporal dynamics are consistent, especially given that pollution levels increased at both locations between sampling points. Nonetheless, caution should be exercised, as observed changes may be influenced by seasonal variations rather than air pollution alone.

Given the current manuscript, it is difficult to convincingly conclude that observed microbiome changes are caused by air pollution rather than inter-individual differences or seasonal effects.

Reviewer #3: (No Response)

Reviewer #4: Thanks to the authors for their hard work. All my questions have been properly addressed and explained.

.

Reviewer #1: No

Reviewer #2: No

Reviewer #3: No

Reviewer #4: No

---

## [Author Response · Author response to Decision Letter 2]

31 Mar 2026

Dear Reviewers,

We sincerely thank you for your careful evaluation of our manuscript and for the constructive comments provided throughout this process. Your feedback has been invaluable in strengthening the clarity and rigor of our work.

In this round, we have provided a detailed, point-by-point response to Reviewer 2, addressing all concerns raised. We hope that our clarifications adequately resolve the remaining questions.

We appreciate your time and thoughtful consideration.

Best regards.

---

## [Editor Report · Decision Letter 2]

9 Apr 2026

Metagenomic and taxonomic profiling of phyllosphere bacteria from Mangifera indica in response to urban air pollutants in Medellín, Colombia.

PONE-D-25-50660R2

Dear Dr. Múnera,

We’re pleased to inform you that your manuscript has been judged scientifically suitable for publication and will be formally accepted for publication once it meets all outstanding technical requirements.

Kind regards,

Massimiliano Cardinale, PhD

Academic Editor

PLOS One
---

## [Editor Report · Acceptance letter]

PONE-D-25-50660R2

PLOS One

Dear Dr. Múnera Porras,

I'm pleased to inform you that your manuscript has been deemed suitable for publication in PLOS One. Congratulations! Your manuscript is now being handed over to our production team.

Kind regards,

on behalf of

Dr. Massimiliano Cardinale

Academic Editor

PLOS One